# Guardians under Siege: Exploring Pollution’s Effects on Human Immunity

**DOI:** 10.3390/ijms25147788

**Published:** 2024-07-16

**Authors:** Gaspare Drago, Noemi Aloi, Silvia Ruggieri, Alessandra Longo, Maria Lia Contrino, Fabio Massimo Contarino, Fabio Cibella, Paolo Colombo, Valeria Longo

**Affiliations:** 1Institute for Biomedical Research and Innovation, National Research Council of Italy (IRIB-CNR), Via Ugo La Malfa 153, 90146 Palermo, Italy; gaspare.drago@irib.cnr.it (G.D.); noemi.aloi@irib.cnr.it (N.A.); silvia.ruggieri@irib.cnr.it (S.R.); alessandra.longo@cnr.it (A.L.); fabio.cibella@irib.cnr.it (F.C.); valeria.longo@irib.cnr.it (V.L.); 2Azienda Sanitaria Provinciale di Siracusa, Corso Gelone 17, 96100 Siracusa, Italy; dipartimento.prevenzionemedico@asp.sr.it (M.L.C.); fabio.contarino@asp.sr.it (F.M.C.)

**Keywords:** environmental pollutant, innate and adaptive immunity, immunomodulation, developmental toxicology, pregnancy

## Abstract

Chemical pollution poses a significant threat to human health, with detrimental effects on various physiological systems, including the respiratory, cardiovascular, mental, and perinatal domains. While the impact of pollution on these systems has been extensively studied, the intricate relationship between chemical pollution and immunity remains a critical area of investigation. The focus of this study is to elucidate the relationship between chemical pollution and human immunity. To accomplish this task, this study presents a comprehensive review that encompasses in vitro, ex vivo, and in vivo studies, shedding light on the ways in which chemical pollution can modulate human immunity. Our aim is to unveil the complex mechanisms by which environmental contaminants compromise the delicate balance of the body’s defense systems going beyond the well-established associations with defense systems and delving into the less-explored link between chemical exposure and various immune disorders, adding urgency to our understanding of the underlying mechanisms and their implications for public health.

## 1. Introduction

The exponential growth of the global economy and the accompanying exploitation of resources and industrial production have caused an unprecedented transfer of contaminants to the environment, with multiple impacts on health. Because these human-induced impacts on earth systems are so extensive, a new definition of this geological era has been proposed, the Anthropocene. Although the last few decades have been characterized by marked technological innovation that aims at the concept of sustainability as a way to improve the quality of the environment, anthropogenic activities continue to have a profound effect on human health, mainly due to poor remediation actions [1]. Humans are regularly exposed to many environmental chemicals that may have potentially toxic effects on health. These chemicals can enter the human body through various routes, including the ingestion of chemicals via the consumption of contaminated food or water, inhalation of airborne pollutants through the respiratory system, and dermal absorption through the skin. 

The most recent WHO environmental burden of disease estimates that every year, 13% of deaths in the 28 European Member States are attributable to environmental stressors [2]. In terms of the absolute number of deaths attributable to the environment, the European Environment Agency concluded that 90% of deaths attributable to the environment result from non-communicable diseases (NCDs), including cancers, cardiovascular disease, diabetes, and chronic lung illnesses (EEA Report No 21/2019). The interplay between the immune system and non-communicable diseases is complex and multifaceted. Chronic inflammation, immune dysregulation, and lifestyle factors can significantly impact the prevention and management of NCDs. Understanding this relationship can provide insights into the prevention and management of NCDs.

The exposome concept was introduced in the field of epidemiology by Wild in 2005 to encompass “the totality of human environmental exposures from conception onwards, complementing the genome” [3]. Therefore, the exposome concept provides a description of lifelong (from the prenatal period) exposure history. It is now recognized that the exposome poses new challenges in assessing the relationship between exposure to environmental factors (pollution, climate change, lifestyle, and diet) and health. Indeed, increasing evidence indicates that genetic variants account for only a limited fraction of the variability in chronic disease risk, as shown by studies in monozygotic twins, leaving a potentially large role for environmental exposures and interactions between environmental and genetic factors [4]. The molecular processes underlying human health and disease are highly complex. Often, genetic and environmental factors contribute to a given disease or phenotype in a non-additive manner, yielding a gene–environment interaction [5]. Epigenetic mechanisms, including DNA methylation, histone modification, and non-coding RNA, can modulate gene expression levels without changing the underlying DNA sequence. Moreover, many epigenetic modifications are dynamic, reflecting cumulative environmental exposures throughout the lifespan and correlating with aging-related diseases and outcomes [6]. 

In 2012, Wild enhanced his approach, describing three overlapping domains within the exposome that refer to different factors, such as: (i) the general external environment (including factors such as the urban–rural environment, climate factors, social capital, and education); (ii) the specific external environment (including diet, physical activity, tobacco, infections, occupation, etc.); (iii) the internal environment, including internal biological factors such as metabolic factors, gut microbioflora, inflammation, oxidative stress, and aging [7]. This, together with the results of studies on animal models, would suggest a pathological mechanism of “multiple hits” [8]. According to this model, the rearrangement of gene expression through epigenetic mechanisms can involve different organ systems and is caused by a combination of genetic predisposition and prenatal injury (the “first hit”), which may not be enough to change the adult phenotype on its own. However, tissue imbalances resulting from the perinatal insults and/or adverse stressors/exposures during postnatal life may serve as a “second hit”, which might reveal or accentuate the underlying abnormalities leading to disease states. 

All chemicals found in the environment are referred to as pollutants or xenobiotics.

Many such chemicals can persist in the environment over long periods of time and, for this reason, are called persistent organic pollutants (POPs). POPs are characterized by ubiquity and persistence in the environment, and it is known that the derived products can interact with the environment and undergo biotransformation and bioaccumulation processes [9]. Among the anthropogenic environmental pollutants, in addition to certain natural compounds such as heavy metals, pesticides, and drugs, synthetic chemical compounds generated during production processes (e.g., hydrocarbons, dioxins, polychlorinated diphenyls, polybromodiphenyl ethers, microplastics, particulate matter, etc.) can contribute to the pathogenesis of several diseases in living organisms.

The immune system is an ancient defense system formed by all metazoans while struggling with various internal and external factors, whose perturbation may lead to increased susceptibility to pathogens and diseases. Inflammation is a common response to a variety of stressors, including xenobiotics. Under normal conditions, inflammation is a healthy and adaptive process that both combats infection and aids in repairing damage to tissues. However, xenobiotics can elicit prolonged, severe, and/or inappropriate inflammatory responses that play a causal role in the progression of biological events, linking a molecular initiating event to an adverse outcome pathway (AOP) [10]. Ordering relevant and causally linked events in the response cascade is critical for applying this information to Environmental Risk Assessment, which is why the AOP framework was created [11].

In this view, the immune system has a recognized central role in many processes involving chronic diseases, and the frontiers between communicable and non-communicable diseases also require additional exploration [12]. The recognition of altered immune system function in many chronic disease states has proven to be pivotal in non-communicable diseases that are characterized by a chronic low level of inflammation, including autoimmune [13] and neurodegenerative conditions [14], cardiovascular diseases [15], cancer [16], diabetes mellitus, and Metabolic Dysfunction-Associated Steatotic Liver Disease (MASLD) [17,18]. Therefore, compared with other toxicity endpoints (i.e., genotoxicity, endocrine toxicity, or developmental toxicity), there are still many challenges facing the introduction of immunotoxicity endpoint evaluation [19]. 

So far, there is still insufficient knowledge about the effects of pollutants on the different cell populations of the immune system, and knowledge of related toxicological mechanisms remains elusive. Many findings have suggested that a subset of environmental pollutants can bind the receptors involved in metabolism, oxidative stress, immune response, and inflammation, such as the aryl hydrocarbon receptors (AhRs) [20]. The complexity of AhR signaling arises from multiple factors, including the diverse ligands that activate the receptor, the expression level of the AhR itself, and its interaction with the AhR nuclear translocator [21]. Indeed, the AhR engages in crosstalk with the AhR repressor or other transcription factors and signaling pathways and, in this way, it can also mediate non-genomic effects [22]. 

In recent decades, the AhR has been increasingly recognized as an important modulator of disease because of its role in regulating the redox system as well as immune and inflammatory responses [23]. The AhR significantly affects the control of adaptive immunity by modulating T cell differentiation and function, both directly and indirectly through its effects on antigen-presenting cells. For instance, 2,3,7,8-tetraclorodibenzo-p-dioxin (TCCD) is a potent xenobiotic ligand of the AhR and was found to inhibit immune responses [24], showing an effect linked to the induction of CD4^+^ T cells with a regulatory phenotype [25]. Indeed, the ability to activate the AhR has been demonstrated for other contaminants, such as polycyclic aromatic hydrocarbons (PAHs) [26] and polybrominated diphenyl ethers (PBDEs) [27]. The production of reactive oxygen species (ROS) and organic metabolites due to the dysregulation of this receptor can cause serious consequences on biological macromolecules such as proteins, lipids, and DNA [28]. Indeed, oxidative stress, inflammation, and apoptosis can compromise immune response, with the possible development of infections, tumors, and lung diseases [29].

In this review, we analyze how environmental pollutants can modulate immune response, reviewing the biological effects of the most common environmental pollutants on different immune cells in in vitro, ex vivo, and in vivo studies. Indeed, we will devote special attention to pregnancy, describing the effects of pollutants on immune response during different windows of prenatal life (Figure 1).

## 2. Pollutants and Immune System Modulation: In Vitro, Ex Vivo, and In Vivo Models

A large set of environmental pollutants have been investigated for their impact on immune response. In this paper, we will review the main biological effects of a list of the most widespread chemicals that have been detected in the environment and whose transfer to humans has been documented. From these perspectives, we will focus on the effects of specific classes of toxicants on the immune regulation of human primary and/or immortalized cell lines and in animal models.

### 2.1. Metals

Heavy metals, such as arsenic, lead, mercury, and cadmium, are a group of highly toxic chemical pollutants that have garnered significant attention due to their detrimental effects on the environment and human health. These elements, which occur naturally in the Earth’s environment, can be released into the environment through various industrial processes, mining activities, and the burning of fossil fuels (anthropogenic activities). One of the most concerning aspects of heavy metals as chemical pollutants is their persistence in the environment. These substances do not degrade over time, accumulating in the soil, water, and air. Consequently, they can enter the food chain, leading to bioaccumulation, where the concentration of these toxic metals increases as one moves up the food web [30,31,32]. Below, we describe the effects of some of the most common heavy metals, analyzing their effects using in vitro, ex vivo, and in vivo models. 

#### 2.1.1. Arsenic

Several national and international organizations have classified Arsenic (As) as the most toxic and carcinogenic inorganic environmental pollutant. It exists in nature in both organic and inorganic forms. The inorganic form is more toxic and accumulates in exposed organisms; it is found in the environment in the form of arsenite (As^III^) or arsenate (As^V^). The main route of arsenic exposure is the consumption of contaminated drinking water. Many studies have found a positive relationship between arsenic (As) exposure and the onset of human pathologies including cancer (lung and bladder) and liver, respiratory, cardiovascular, and immune diseases [33,34]. In terms of immune system dysregulation, it has been demonstrated that As either reduces the expression level of cytokines (IFN-γ, IL-4, and IL-10) [35,36,37] or induces apoptotic mechanisms in different cell types, such as B and T lymphocytes, macrophages, and neutrophils [38,39]. It has been shown that As can drive M2 polarization in macrophages, triggering the CD206 marker along with TGF-β1 and Arg1 genes that have been shown to be involved in the fibrinogenic process [40]. Arsenic-induced M2 polarization is also favored via miR-21 regulation of phosphatase and tensin homolog deletion on chromosome ten (PTEN), which has been implicated in chronic hepatic fibrosis and in the development of arsenicosis [41].

Microarray analysis of peripheral blood mononuclear cells (PBMCs) purified from subjects exposed to increasing As concentrations has also shown that this pollutant modulates T cell receptor expression, the cell cycle, and apoptotic processes. Furthermore, As increases the expression of inflammatory molecules such as cytokines or growth factors [42] and induces the hypomethylation of leukocyte DNA, which is associated with an increased risk of inflammatory skin lesions [43]. Arsenic also has a negative effect on innate and humoral immunity, altering macrophage and B lymphocyte functions and stimulating oxidative stress signaling, DNA damage, and cytotoxicity in T cells and in human polymorphonuclear neutrophils [44,45,46]. 

#### 2.1.2. Lead

Lead (Pb) is available in nature in relatively low amounts. Anthropogenic activities, such as manufacturing, mining, and the burning of fossil fuels, have contributed to the increased release of Pb into the environment. Lead induces toxic effects on various human tissues, including the immune system, leading to increased allergies and infectious and autoimmune diseases [47,48]. For instance, in vitro studies have shown that Pb treatments induce the dysregulation of pro-inflammatory cytokine production [49,50,51] and impair THP-1 monocyte/macrophage cell viability at low concentrations. Zhao and coworkers showed that Pb exposure induces the quiescence of hematopoietic stem cells through Wnt3a/β-catenin signaling and decreases the expression of CD70 on bone marrow-resident macrophages [52]. These observations were confirmed by ex vivo studies on a population of chronically exposed workers, in whom higher levels of IFN-γ, IL-2, IL-12, and IL-17 were observed [53,54]. Lead can cross the placenta [55] and can have adverse effects on birth outcomes, possibly by accumulating in the placenta and causing reduced nutrient transfer, oxidative stress, and abnormal functioning [56]. 

#### 2.1.3. Mercury 

Mercury (Hg) is a highly toxic metal, and exposure to its organic and inorganic chemical compounds causes adverse effects on humans, inducing genetic damage and neurological, kidney, cardiac, and immunological diseases [57]. The predominant toxic Hg forms include elemental Hg (Hg0), ionic Hg, and organic (o)Hg, such as methylmercury (MeHg), which is classified as the most toxic among them.

In the innate immune system, Hg can influence the activity of different cell subtypes, such as neutrophils, monocytes/macrophages, NK cells, and dendritic cells. It has been shown that in vitro exposure of neutrophils to low concentrations (≤5 μM) of HgCl_2_ causes an increase in the production of superoxide ions [58], inhibiting the apoptosis of neutrophils, while at higher concentrations, Hg displays cytotoxic effects. At the same concentrations of pollutant, macrophages show functional impairment of both phagocytic and migratory activity [59,60], showing a reduction in NO production and a greater synthesis of inflammatory cytokines, such as IL-6 and TNF-α. Moreover, after Hg exposure in co-culture assays of intestinal epithelial cells and THP-1 macrophages, mimicking the gastrointestinal tract, the metal induced a pro-inflammatory response with an Increase in IL-8 and IL-1β cytokine production [61]. 

Mercury derivatives also show genotoxic effects in vitro and in vivo, in terms of both chromosome aberrations and loss of cell proliferative capacity, compromising the metabolic, immune, and nervous systems [62]. Dietary MeHg intake (3.9 μg/g) by mice and rats resulted in a 42–44% suppression of the tumoricidal activity of blood and splenic NK cells, as well as the proliferation of T and B cells of the adaptive immune system [63,64]. These events have been associated with an increase in intracellular oxidative stress [65,66], which causes the activation of transcription factors and the expression of genes encoding pro-inflammatory cytokines. 

#### 2.1.4. Cadmium

Cadmium (Cd) is a heavy metal naturally present in the environment (in the soil, water, and air) and is mainly used in the steel industry, in plastics, and as a component of batteries. It is released into the environment through industrial and domestic activities, such as the combustion of fossil fuels (coal, diesel, gasoline, etc.), the incineration of industrial waste (particularly batteries and plastics containing Cd), the production of metal alloys, and the manufacture of fertilizer phosphates. Food, drinking water, and the inhalation of cigarette smoke, in both active and passive forms, are sources of this heavy metal. Target organs for Cd-induced toxicity include the liver, kidneys, lungs, testes, prostate, heart, skeletal system, and the nervous and immune systems.

Following environmental and/or industrial exposure, it has been shown that Cd induces inflammation and oxidative damage in neutrophils and macrophages, leading to an increase in the production of ROS [67,68]. These results have been further confirmed by experimental studies carried out on macrophage cell lines [69]. Cd-induced M1 polarization has been observed in in vitro and in vivo rodent macrophages via the JAK2/STAT3 pathway, contributing to the atherosclerotic process [70]. Furthermore, it has been shown that intrauterine exposure to Cd can provoke a dramatic decrease in the levels of IFN-γ and IL-2 cytokines in offspring and affect the activity of adaptive immunity cell populations in C57Bl/6 mice. In female offspring, CD4^+^ and CD8^+^ T cell populations increase after exposure to Cd, while the percentage of NK cells and granulocytes decreases. In addition, a decrease in splenic Treg cells has been shown in both male and female offspring [71,72]. Another study in wild boars highlighted that Cd downregulates the expression of different pro-inflammatory cytokines (TNF-α, IL-12p40, several TLRs, CD14, MD2, BD2, MyD88, p65, and NOS2) in macrophages, negatively affecting their roles in immunity [70].

In a study on workers occupationally exposed to Cd, epigenetic alterations in the expression of miRNAs associated with inflammation and carcinogenesis, as well as their possible correlation with the immune profile, were evaluated. MiRNA profiling showed that miR-221 and miR-155 were modulated in exposed workers with reference to a higher percentage of the Th17 population [73] (See Table 1 for summary).

### 2.2. Persistent Organic Pollutants (POPs)

Among the most dangerous organic pollutants for health, persistent polyhalogenated organic compounds or POPs (so-called because they withstand biochemical and photolysis processes by remaining in the environment for a long time) [74] are very stable chemicals whose lipophilic nature determines their bioaccumulation in animal adipose tissues, thus allowing their biomagnification along the food chain. Due to their chemical-physical characteristics, POPs can cross the placenta, causing exposure starting from intrauterine life, but can also accumulate in breast milk, which is particularly rich in lipids, determining their intake in post-natal life [75]. The most common POPs are polycyclic aromatic hydrocarbons (PHAs), organochlorine pesticides (OCPs), polychlorinated biphenyls (PCBs), perfluorinated compounds (PCFs), dioxins, and brominated flame retardants (BFRs), which will be discussed in their dedicated sections. 

#### 2.2.1. Dioxins

Dioxins are highly toxic POPs produced from natural processes such as forest fires as well as the burning of trash by anthropic activities. Due to their highly lipophilic nature, they undergo biomagnification along the food chain to eventually accumulate in human tissues. Of all the analyzed congeners, 2,3,7,8-tetrachlorodibenzo-p-dioxin (TCDD) showed the highest toxicity for living organisms [76]. Several in vitro, ex vivo, and in vivo studies have been conducted to determine the adverse effects of dioxins on the various components of the immune system. Regarding the innate response, a compromise in the defense system against pathogenic organisms and cancer cells was observed; in particular, TCDD impairs the correct functioning of macrophages, NK cells, neutrophils, and dendritic cells [77,78,79]. In adaptive response, dioxin modulates the production of antibodies by B lymphocytes and the cytotoxic activity of T lymphocytes, while also attenuating the IgE-mediated hypersensitivity response [80,81]. It has been observed that, in THP-1-derived macrophages, TCDD treatment induces alterations in adherence, adhesion molecule expression, morphology, multiple cytokine/chemokine production, and the expression of total mRNA [82]. Few studies have focused on TCDD’s multigenerational and transgenerational effects on human reproductive health, despite the large amount of evidence in animal models of such effects on male and female reproductive health [83]. 

#### 2.2.2. Polychlorinated Biphenyls (PCBs)

Polychlorinated biphenyls are environmental pollutants of industrial origin. Although their production was interrupted many years ago, they persist in the environment due to their resistance to chemical, physical, and biological degradation processes. Like other organic pollutants, they accumulate in the soil and can enter the food chain, eventually reaching humans. Numerous studies have shown that PCBs can alter the proper functioning of the immune system at several levels. For example, in macrophage cell lines, treatment with PCB 126, a dioxin-like polybromophenyl, has a pro-inflammatory immunostimulating effect [84]. In vitro, the non-dioxin-like polychlorinated biphenyl (NDL-PCB) congeners, NDL-PCB 153 and NDL-PCB 180, display an immunosuppressive effect by reducing the expression of the pro-inflammatory cytokines TNF-α and IL-6 and the expression of reactive species [85]. PCBs are also capable of altering the physiological antiviral cellular response [86]. In an in vivo mouse model, PCB promoted alterations in collagen in the female bladder compared to the male, indicating an inflammatory event that impairs immune system functions [87]. Moreover, studies from PCB-exposed populations have revealed DNA methylation differences in CpG sites with effects on estradiol CpG site sizes and the immune system [88]. 

#### 2.2.3. Brominated Flame Retardants (BFRs)

Brominated Flame Retardants (BFRs) are chemicals used in the synthesis of plastics, textile products, and electronic and electrical devices; they are added to industrial products to make them less flammable. It has been shown that the flame retardants contained in various products are released into the environment through combustion, percolation, and degradation [89]. Due to their increasing use, BFRs have become globally widespread pollutants over the years. They have been found both at the level of different environmental matrices (air, aquatic sediments, soil, dust, etc.) and in animal organs and tissues, such as blood, mother’s milk, and adipose tissue [90,91].

There are four main classes of BFRs: (1) polybrominated diphenyl ethers (PBDEs), present in plastics, textiles, electronic devices, and circuits; (2) hexabromocyclododecanes (HBCDDs) used for thermal insulation in buildings; (3) tetrabromobisphenol A (TBBPA) and other phenols used in thermoplastics and televisions; (4) polybrominated biphenyls (PBBs) used in consumer devices and textiles [92]. The immunotoxic effects of the most studied flame retardants will be discussed below. 

##### Polybrominated Diphenyl Ethers (PBDEs)

Polybrominated diphenyl ethers are chemicals consisting of two aromatic rings joined by an ester bond with bromine atoms. The commercial use of PBDEs dates back to 1976 and the three most used blends were penta-BDE, octa-BDE, and deca-BDE. Since 2004, the production of penta- and octa-BDE has been banned, both in the United States and in the European Union, due to their negative effects on the environment and health. The deca-BDEs have been banned in the United States since 2010. On the basis of the number and position of the bromine atoms, we can distinguish 209 congeners with different chemical-physical characteristics that affect their capacity for bioaccumulation and toxicity in the liver, kidney, gut, and thyroid in terms of oxidative and mitochondrial damage and apoptosis [90,93]. Human population studies have shown that PBDEs accumulate in adipose tissue, hair, human serum, breast milk, and the placenta. Among the most toxic and widespread congeners in the environment, BDE-47 (a tetra-BDE) and BDE-209 (a deca-BDE) are the most relevant in terms of health risks. Experimental data on animal models have shown that BDE-47 can have neurotoxic, cardiotoxic, hepatotoxic, and teratogenic effects on zebrafish larvae and adults and in fish [94,95,96].

Several studies have investigated the immunotoxic effects of PBDEs. In in vitro studies, BDE-47 significantly reduced the expression and secretion of the pro-inflammatory response M1 macrophage marker genes IL-6, TNF-α, and IL-1β and the expression of metalloelastase 12, a gene essential in the motility of the macrophage, recruitment of neutrophils, and release of cytokines and chemokines in inflammatory processes [97]. Indeed, the same authors showed that BDE-47 can have diverse action mechanisms on human macrophage cell lines, affecting small extracellular vesicle biogenesis [98] and the content of their miRNA cargo. Furthermore, they demonstrated that BDE-47 treatment can regulate the sEVs’ cargo with purposeful consequences toward downstream events and target bystander macrophages, exacerbating the macrophage LPS-induced pro-inflammatory response [99] as well as altering A549 epithelial lung cells, which modulate the mRNA expression of tight junctions, adhesion molecules, cytokines, and epithelial–mesenchymal transition markers [100]. Finally, it has been shown that BDE-47 can also induce cardiovascular toxicity, activating PPARγ in THP-1 monocytes and inducing the foam cell formation typical of a proatherogenic process [101]. In an animal model, it was observed that they stimulate an increase in the production of ROS in macrophages, leading to the remodeling of the cellular physiology of phagocytes and, therefore, to the destruction of the functional activity of the macrophage [102,103,104]. Recently, Balb/c mice treated with different doses of BDE-47 by gavage showed that BDE-47 significantly reduced antibody response and induced histopathological effects on the liver, spleen, small intestine, and thyroid in the absence of general toxicity, suggesting that exposure to BDE-47 may perturb innate and adaptive immune responses [105].

##### Hexabromocyclododecane (HBCDD) and Tetrabromobisphenol A (TBBPA)

Hexabromocyclododecane (HBCDD) is a non-aromatic brominated cyclic alkane; it is mainly used in expanded propylene insulation foam [106,107], but also in tapestry fabrics, curtains, and wall coverings [108]. HBCDD is added to plastic material but, since it is not chemically linked to it, it can transfer into the environment and accumulate in the soil, sediments, and dust particles. It is involved in the trophic chain to humans through the ingestion of dust [109,110], bioaccumulating in the blood, adipose tissue, and breast milk [111,112,113]. Immunotoxicity studies on rats have highlighted that HBCDD reduces splenocytes and increases IgG and neutrophils, contributing to immune system imbalance [114]. Tetrabromobisphenol A (TBBPA) is a flame retardant used in thermoplastics production. At the cellular level, Barańska and collaborators found that TBBPA has a genotoxic mechanism because it causes oxidative damage in purines and pyrimidines in PBMCs [114,115,116]. Furthermore, a study conducted on human dendritic cells isolated from healthy subjects by Canbaz and collaborators showed that exposure to this flame retardant can induce an inflammatory phenotype typical of a Th2 response [117]. Furthermore, for the same substance, neurotoxic, nephrotoxic, hepatotoxic, and immunotoxic effects have been reported, which can induce ROS expression, promoting the production of inflammatory cytokines and compromising mitochondrial function [118]. A correlation has also been demonstrated between human exposure to both classes of pollutants and thyroid and neurological disorders, reproductive health, and immunological, oncological, and cardiovascular diseases [119,120].

#### 2.2.4. Bisphenol A

Bisphenol A (BPA) is a chemical that has been synthesized since the 1960s and is widely used in industrialized countries. BPA is used in the production of polycarbonate plastics, in food-grade containers, and in epoxy resins that make up the internal protective coating found in most food and beverage cans. In addition, it is also used for dental devices and in thermal receipt paper. Under the pressure of EFSA, between 2010 and 2015, many studies aimed at understanding the possible effects of BPA on immune system cells and tissues were conducted. Balistreri and collaborators showed that BPA alters the function of neutrophils in a dose-dependent manner, causing an increase in ROS production in these cells through a calcium-dependent process. Transwell chemotaxis assays revealed that BPA exposure reduces the chemotactic capacity of neutrophils in a gradient of the bacterial cell wall component f-Met-Leu-Phe, a potent chemoattractant. Indeed, exposure to BPA also inhibits the ability of neutrophils to kill methicillin-resistant Staphylococcus aureus, a leading human pathogen. These data suggest that BPA could compromise some fundamental functions of the innate immune response [121]. Bisphenol A also promotes the polarization of M1 macrophages In terms of cell number, the gene expression of CD11c and iNOS markers, and the production of pro-inflammatory cytokines via the upregulation of transcription factor IRF5 expression, which is involved in the polarization process [122]. The upregulation of pro-inflammatory cytokines was also induced in the murine macrophage cell line RAW264.7 by the BPA analog BPF (4,4′-Methylenediphenol). Together with the involvement of the JAK2/STAT3 pathway, it contributed to M1 polarization [123,124].

BPA also showed immunotoxic effects on the adaptive immune response. An in vivo study conducted on mice showed that animals treated with a single dose (250 µg/kg) of BPA in the neonatal period and subsequently subjected to breast cancer induction had no significant changes in the population of B lymphocytes or damage to the spleen and peripheral lymph nodes. However, they exhibited a significant difference in IgM reactivity, with a notable decrease in the ability to recognize tumor antigens in guinea pigs [125]. 

#### 2.2.5. Perfluorooctanesulfonic Acid and Perfluoroctanoic Acid 

Perfluorooctanesulfonic acid (PFOS) and perfluorooctanoic acid (PFOA) are perfluoroalkyl organic substances (PFAS). They are chemical compounds containing long carbon chains that are used to increase resistance to high temperatures in numerous products, such as fabrics, carpets, clothing, food-grade paper, non-stick cookware, and firefighting foams. PFOA and PFOS remain in the environment for a long time and, due to their chemical stability, resist degradation processes, accumulating in the soil, air, and water. PFAS are able to provoke the release of the pro-inflammatory cytokines IL-1β and caspase-1 in THP-1 macrophages. They also activate the innate immune response through the AIM2 receptor of inflammasomes, inducing inflammation in the lung, liver, and kidneys of mice [126]. Population studies have shown that PFAS can accumulate in human organs and tissues; for example, PFOS has been found in the liver, kidneys, lungs, hair, breast milk, and urine, but it accumulates predominantly in the blood [127,128]. In vitro and in vivo data have shown that PFOSs and PFOAs have toxic effects on multiple cell types and organs of the immune system. An in vitro study showed that treatment of human T cells with increasing concentrations of PFOS induced a reduction in the expression of IL-2, a cytokine essential for the correct functioning of leukocytes and whose reduction was observed in various autoimmune diseases [129]. An in vivo study conducted in mice showed that PFOS treatment inhibits T cell proliferation. Analysis of the biochemical pathways involved in immune cell signaling revealed the inhibition of genes involved in cell cycle regulation and response to oxidative stress [104]. In vivo studies in mice performed by Torres and collaborators showed that PFOS exposure reduced immune cell populations in some organs and also led to an increase in the number of cells in others, suggesting a possible de novo localization of cells and a decrease in the activity of some organs such as the spleen and liver [130]. Taylor and coworkers showed that PFOA exposure can reduce B-cell subtypes and, consequently, the IgM antibody primary response in female mice [131]. 

### 2.3. Volatile Organic Compounds (VOCs)

Volatile organic compounds are a large group of organic chemical compounds that consist of elements of carbon and one or more functional groups containing oxygen, phosphorus, nitrogen, silicon, halogen, and sulfur. VOCs are characterized by high vapor pressure and low water solubility. These molecules are produced by anthropogenic activities such as the manufacture of paints, solvents used in furniture making and electronics, vehicular traffic, refrigerants, and pharmaceuticals. They are also emitted during natural processes, including forest and vegetation fires, volcanic eruptions, and microbial processes [132]. VOCs show high resistance to abiotic and biotic degradation, and by interacting with environmental pollutants (i.e., nitrogen oxide species), they produce tropospheric ozone and secondary environmental pollutants which have hazardous effects on the well-being of humans and ecosystems [132]. Since the 1990s, the relationship between VOC exposure and immune response has been extensively studied, particularly in murine and ex vivo human models. Multiple data sources have shown the ability of different types of compounds to induce an imbalance of reactive oxygen species and the release of pro-inflammatory cytokines from immune cells by activating the inflammatory molecular pathways of NF-kB. Studies have shown that exposing CD3+/CD28+ human PBMCs to toluene decreases the release of IL-4 and IL-13 cytokines while simultaneously increasing the production of the pro-inflammatory cytokine TNF-α [133]. Xylene, another aromatic hydrocarbon, induces mitochondrial damage following oxidative stress in purified human T lymphocytes [134]. Genome-wide analyses have demonstrated that VOCs induce the activation of genes belonging to type I Interferon signaling in human promyelocytic leukemia HL60 cells [135]. A cross-sectional study of workers exposed to formaldehyde suggested a reduction in NK cells, regulatory T cells, and CD8^+^ effector memory T cells compared to control workers [136]. The aforementioned studies, along with many others related to the effect of different types of volatile organic compounds on the modulation of the immune response, have been thoroughly reviewed by Ogbodo and coworkers in a recently published paper [137], where the authors have also linked the inflammatory response to VOCs to the onset of several autoimmune diseases.

### 2.4. Particulate Matter (PM)

Airborne particulate matter may be a complex mixture of dust, dirt, soot, and smoke containing both organic and inorganic substances as well as biological components. The airborne particulates can be characterized by their physical attributes, which influence their transport and deposition, and their chemical composition, which influences their effect on health and the immune system. The particles most likely to cause adverse health effects are the fine particulates PM_10_ and PM_2.5_ (particles smaller than 10 microns and 2.5 microns in aerodynamic diameter, respectively). PMs play a role in biological pathways related to cell signaling and immune response [138]. Even if the effects of short-term exposure to PM_2.5_ on innate immunity and its relationship with tissue damage and repair ability have been scarcely studied [139], it has been demonstrated that prolonged exposure to PMs, even at low levels, is associated with the activation of inflammatory pathways (e.g., Nrf2, MAPK, and NF-kB), with an increase in cytokine and chemokine production [140] and immune cell activation [141]. PM can contribute to the development and maintenance of inflammation through epigenetic mechanisms, such as changes in histone tail modifications, microRNA (miRNA) expression, and DNA methylation [141].

By employing ALI (Air Liquid Interface) cultures of various human cells that represent different sections of the respiratory tract, researchers have successfully simulated realistic exposure conditions to airborne particles. This approach has enabled the investigation of toxicity and immune responses following controlled exposure on the first line of immune responder cells. These models are essential for evaluating the effects of PM, including oxidative stress, inflammation, DNA damage, and apoptosis [142]. Recently, Chivé et al. reported that exposure of the Calu-3 cell line to ultrafine particulates in the ALI exposure system induces a pro-inflammatory response and reduces antiviral defenses. Their findings may elucidate the observed association between PM_2.5_ exposure and the severity of influenza infections in human populations [143].

Alternatively, in vitro studies on the effects of PM exposure on immune cells can be conducted by treating cells with resuspended atmospheric particulates in culture media. In particular, research by Danielsen and collaborators revealed that PM exposure has similar effects on lung epithelial cells (A549) and monocyte cultures (THP-1). The response varied based on the particulate collection site, with the most significant effects observed in areas with high biomass combustion [144]. Marín-Palma and collaborators identified the differential expression of 1196 genes following the exposure of PBMC cells to PM_10_. The differentially expressed genes related mainly to the inflammatory response, with an upregulation of cytokines and chemokines. Interestingly, this alteration in the inflammatory response was accompanied by the downregulation of genes involved in pathogen defense and antiviral factors [140]. Moreover, an in vivo study on a rat model reported that short-term exposure to high concentrations of PM_2.5_ induces lung tissue damage and alters innate immune responses. Interestingly, although some repair occurs after 15 days of recovery, the function of innate immunity, including immune defense and lung homeostasis regulation, remains partially impaired [139]. 

In a study using a rat model, Wang and colleagues found that exposure to traffic-related PM_2.5_ impairs regulatory T cell (Treg) function and disrupts the balance between T-helper 1 (Th1) and T-helper 2 (Th2) cells. This disruption occurs through differential microRNA (miRNA) expression and DNA methylation modifications, ultimately leading to the exacerbation of asthma symptoms [145].

### 2.5. Microplastics

Microplastics (MPs) are small plastic particles with diameters less than 1 mm and have become a pervasive environmental pollutant. Originating from the breakdown of larger plastic debris, as well as from products such as cosmetics, textiles, and industrial processes, microplastics become part of the trophic chain in both marine and terrestrial ecosystems, posing a threat to human health [146,147]. So far, MPs have been detected in different biological fluids and cellular compartments as milk, blood, animal excreta, and placenta [148,149]. Microplastics vary significantly in their chemical composition, size, and shape. These variations can affect exposure levels, local or systemic absorption, and interactions at cellular and molecular levels with the plastics [150]. Polystyrene (PS), Polyethylene (PE), Polypropylene (PP), and Polyvinyl chloride (PVC) are among the most frequently found plastics in the environment [151]. These MPs are hydrophobic, making hydrophobicity a key factor in microplastics’ role as vectors for a range of contaminants. It has been observed that microplastics can cause damage to cells in several ways, inducing oxidative stress, inflammation, metabolism disorders, apoptosis, and genotoxicity. PS MPs affect immune function in microglial HMC-3 cells by depositing onto brain microglial cells, altering their cellular morphology and causing immune dysfunction through the alteration of IL-1β, CCL2, and TGF-β levels [152]. In vitro studies using sea bass or seabream fish seem to suggest that the long-term exposure of fish to PVC or PE microplastics could impair fish immune parameters, probably due to the oxidative stress produced in their leucocytes [153]. In addition, in vivo experiments on chickens have demonstrated that exposure to PS MPs causes liver damage and inflammation, autophagy activation, intestinal macrophage infiltration, and lipid accumulation [154]. Additionally, polypropylene (PP) particles induce pro-inflammatory cytokines in a size- and concentration-dependent manner by increasing the levels of cytokines and histamines in PBMCs and RAW 264.7 and HMC-1 cells. Studies on PBMC and THP-1 macrophage-like cell lines have been conducted to investigate metabolic activity, cytotoxicity, ROS production, and macrophage polarization [155]. In addition, it has been shown that UV radiation accelerates the photo-oxidation and aging process of plastics, leading to changes in their chemical structure and physical properties. This process not only contributes to the fragmentation of plastics but also affects their morphology and the release of harmful additives and chemicals, further contaminating ecosystems. Recently, it has been shown that treatment with photoaged PS MPs induced increased malondialdehyde levels [156] in the THP-1 macrophage-like cell line. 

Table 2 summarizes the effects of POPs, VOCs, PMs, and MPs on immune response.

Figure 2 displays the similarities and differences between the main classes of environmental pollutants in the modulation of immune response.

## 3. Developmental Immunotoxicology

The term developmental immunotoxicology (DIT) has been used to identify a new approach to understanding the effects of immune development perturbation on adverse outcomes [157]. DIT addresses the impact of xenobiotics on immune responses during the early stages of immune system development/maturation compared to adults. Immunotoxicology studies have defined some critical windows of vulnerability during development [158], when xenobiotics can exert their effects, but sometimes stressors may be applied during rather broad periods which often cover the entire immune developmental period. These effects can be particularly significant in highly polluted areas where environmental stressors are prevalent throughout a large portion of prenatal and postnatal life. Furthermore, considering the accumulation of evidence supporting the hypothesis of the Developmental Origin of Health and Disease (DOHaD) [159,160], disturbances of the immune system during fetal development are of particular interest for possible long-term consequences. According to epidemiological studies, the adverse effects of in utero exposures on the immune system have been associated with higher rates of chronic immune disorders in adults, including autoimmunity, immune deficiency, inflammation, and allergic reactions [161].

Although these associations are of considerable importance, only examining outcomes directly linked to immune system elements fails to capture the complete range of health risks associated with DIT. In fact, from the early stages of life, immune cells are found in all tissues and organs where they serve as homeostatic regulators of the physiological function of the tissue and as sentinels to defend it. In the liver, for example, the functional state of hepatocytes (the majority of cell populations) can be significantly altered by a small number of resident immune cells (i.e., Kupffer cells). This occurs similarly between microglia and astrocytes in the central nervous system and many other organs [162]. For these reasons, the involvement of DIT events underlying behavioral disorders and metabolic dysfunction has been suggested [163].

During pregnancy, major adaptations occur in the maternal immune system to protect the mother and her future baby from pathogens. The perturbation of this balance may have detrimental immune responses against the allogeneic fetus. Pro- and anti-inflammatory stimuli alternate during gestation, and their regulation is the basis of the success of numerous key steps of the pregnancy. For example, the implantation of the embryo on the uterine wall is facilitated by a period of active inflammation, essential for the remodeling of the maternal uterus [164]. On the contrary, the phases of active fetal growth and development are accompanied by a relative inflammatory quiescence [165]. Failure to induce these systemic changes predisposes women to adverse pregnancy outcomes, such as recurrent miscarriage [166], preeclampsia [167], and preterm birth [168]. Additionally, depending on the duration and severity, this inappropriate activation of the maternal immune response may have transgenerational consequences. For instance, the activation of the mother’s immune/inflammatory response has been linked to negative neurobehavioral outcomes in offspring [169,170,171,172]. These findings are noteworthy from an immunological standpoint, as the physiological development of the fetal brain involves a variety of immunological factors at the maternal–fetal interface [173]. For example, in the case of extremely preterm infants, placental methylation of genes related to inflammation has been observed to be associated with a decreased risk of cognitive impairment and decreased neonatal systemic inflammation [174]. 

Evidence suggests that prenatal exposure to xenobiotics may play a role in altering the epigenetic regulation of immune-related genes that influence the development of regulatory T cell (Treg) networks or the ratio of Th1 and Th2 cells. Because Th1/Th2 immunity is closely linked to disease, epigenetic changes caused by an adverse intrauterine environment could explain the occurrence of disease susceptibility later in life [175]. In parallel with the direct modulation of epigenetic regulation, numerous toxic substances can impact the endocrine system. Given its close collaboration with the immune system in guiding development from gestation to early childhood, endocrine disruptor chemicals may also trigger DIT [176].

In this section, we will provide an overview of the effects of the most relevant xenobiotics during pregnancy, with specific reference to immune dysregulation.

Arsenic toxicity has been linked to many possible pathways, including oxidative stress, epigenetic modification, interference with DNA repair mechanisms, and alterations of the immune system [177,178]. Exposure to As during pregnancy can result in DIT interacting directly and indirectly with the fetal environment. This interference is strongly related to placental function, as the placenta is permeable to As and particularly susceptible to its detrimental effects, including oxidative stress, inflammation, and immune system perturbation [33]. In two separate cohort studies from Bangladesh and the United States, prenatal As exposure was associated with lower percentages of CD4^+^ T cells and, in particular, activated memory CD4^+^ T helper cells in cord blood [179,180]. Although a direct comparison of exposure levels between the two populations is not possible, the effects of As exposure appear to be consistent. In the study on the Bangladeshi population, As exposure is determined from the levels found in drinking water samples, whereas in the U.S. study, exposure is assessed from maternal toenail samples. Evidence of the similarity in findings showed that both cohorts demonstrated relatively higher susceptibility to infectious diseases at different stages of childhood, which was associated with prenatal arsenic (As) exposure [181,182]. This suggests that prenatal As exposure causes overall immunosuppression in offspring. Among the mechanisms of action that could explain the negative effects of As exposure on immune system cells, the most widely accepted appears to be the induction of oxidative stress. Arsenic exposure has been shown to trigger oxidative stress in peripheral blood in both adults and children [183,184]. Studies conducted on populations exposed to high levels of As have demonstrated that prenatal exposure to arsenic increases the expression of numerous markers of inflammatory and oxidative stress in umbilical cord blood and placenta [33,185].

The long-term effects of arsenic exposure during pregnancy on the immune system may be mediated by epigenetic mechanisms. Specifically, exposure to arsenic has been shown to induce alterations in DNA methylation patterns, which can influence gene expression and immune cell function [186,187]. Specifically, the authors identified 10 differentially methylated CpGs corresponding to six genes (BAI2, CD1B, TEAD1, SGCZ, ACSS3, and ZMYND19) associated with Gestational Diabetes Mellitus with high Arsenic (As) exposure. Studies on mice have suggested that one possible mechanism linking prenatal As exposure to health effects later in life may be epigenome alteration. Cytosine methylation (e.g., CpG methylation) has been suggested as a valuable candidate for As-related health effects [188].

Toxic effects related to Hg exposure, particularly methylmercury (MetHg), mainly affect the nervous system [189]. However, as previously mentioned, in vitro experimental evidence showed possible associations between Hg exposure and immunotoxicity [61,65,67]. Mercury can be transferred from mother to fetus through the placenta [189], and in utero exposure to Hg has been associated with a higher risk of respiratory infection in infants during the first year of life [190]. Mercury levels during pregnancy and postpartum have been linked to the frequency of natural Treg and NKT cells, as shown in the New Hampshire Birth Cohort Study. Higher natural Treg counts have also been associated with an increased risk of Hg exposure as measured by seafood consumption. Elevated Tregs may represent a counterbalancing mechanism in response to the increased autoimmune pressure associated with low-level Hg exposure [191]. In a study focused on a highly exposed population from the Faroe Islands, Oulhote and colleagues reported a correlation between prenatal methylmercury (MetHg) exposure and a decreased total white blood cell count, specifically affecting monocytes, basophils, and CD3^+^ T, CD4^+^ T, and CD19^+^ B lymphocytes at the age of 5 [192]. It could be hypothesized that such an alteration in the balance of white blood cells could affect inflammatory cytokine levels. In this view, a multicentric cohort study conducted in five European countries (France, Greece, Norway, Spain, and the United Kingdom) showed that in utero exposure to Hg was correlated with poorer childhood metabolic status and higher levels of TNF-α, IL-6, and IL-1 at 6 and 12 years. In this report, the authors suggested that changes in key inflammatory cytokines and the metabolic profile are strictly interconnected [193]. In an Amazonian Brazil population exposed to high levels of mercury, a positive association between mercury levels and serum concentrations of antinuclear antibodies was found, suggesting a possible role of mercury in autoimmune disease [194]. In a previous study conducted in the same population, Pinheiro and colleagues reported elevated glutathione levels and decreased catalase activity in response to Hg exposure [195]. Mercury’s capacity to modulate the synthesis and activity of enzymes within the endogenous antioxidant system may underlie its effects on immune cells. By disrupting the normal function of these enzymes, mercury can impair the antioxidant defenses of immune cells, resulting in altered immune responses and increased vulnerability to infections and diseases.

Studies on the effects of Pb during pregnancy suggest that exposure to the metal can lead to adverse outcomes, such as an increase in enzymes involved in oxidative stress, tissue damage, and the dysregulation of inflammatory pathways [196], and a polarization towards a Th2-type response with elevated serum IgE levels [51,197]. Indeed, lead exposure can affect thymic function, altering the development and maturation of immune cells [198]. This can cause an imbalance in immune functionality, shifting towards a more pro-inflammatory response and increasing the susceptibility to allergies and autoimmune diseases. In utero Pb exposure has been identified as a risk factor for childhood asthma later in life [47]. Indeed, Pb-related immunomodulating effects have also been highlighted by cross-sectional studies that revealed associations between blood Pb levels and biomarkers linked to allergy and infectious diseases in children [199,200]. 

Cadmium exposure induces oxidative stress by generating ROS, through which it exerts its toxicity. Cd exposure can occur through the consumption of contaminated food as well as through the inhalation of cigarette smoke. Pregnant women who smoke have considerably greater amounts of Cd in their blood, placenta, and umbilical cord blood. Cd concentration was associated with a higher expression of miRNA in cord plasma [201]. Cd exposure was associated with respiratory symptoms in adolescents. One proposed mechanism of action is increased vulnerability to acute respiratory infections in the early years of life [202,203]. The accumulation of Cd in the placenta has been proven in both in vitro and epidemiological studies [204,205]. Cadmium concentrations have been analyzed in cord blood, maternal blood, and placental tissue, with demonstrated Cd-induced oxidative stress that adversely affects birth outcomes [206]. Sanders and coworkers found a significant association between in utero Cd exposure and DNA methylation patterns in the leukocytes of newborns and their mothers. It is worth noting that the methylation pattern was Cd-specific; in fact, when compared to those produced using cotinine rather than Cd levels, the methylation patterns were non-overlapping [207]. Cadmium levels in the urine of pregnant women were associated with a lower absolute number of CD3^+^ and CD4^+^ lymphocytes and lower levels of IL-4 and IL-6 in girls, while an Inverse association with the absolute count of CD3^+^ and CD8^+^ was only seen in males, according to a recent cohort study done in Wuhan, China [208]. Similarly, a sex-related effect was noted in a study conducted by Nygaard and colleagues. They reported that the concentration of Cd in maternal nails, sampled immediately after delivery, was associated with a decreased number of T helper memory cells in cord blood [180]. Moreover, studies in rats have demonstrated that the sex-specific effects of cadmium on immune system cells might result from its interaction with sexual hormones, particularly 17β-estradiol [209]. 

Exposure to PFAS and PFOA has been associated with the reduced immunogenicity of vaccines in offspring in two independent cohort studies from Norway and the Faroe Islands [210,211,212]. In the cohort of Faroese children, prenatal exposure to MetHg, PCBs, and PFASs was also associated with higher levels of autoantibodies [213]. However, the authors concluded that more comprehensive cohort studies are needed to determine whether the presence and concentrations of these autoantibodies can predict the occurrence and severity of autoimmune diseases, or whether their presence should be considered merely as a consequence of tissue damage. 

PFAS exposure during pregnancy has also been linked to increased risks of asthma and respiratory syncytial virus infection in childhood [214]. Along the same lines, a study conducted using the National Health and Nutrition Examination Survey (NHANES) survey reported an increased risk of asthma in adolescents exposed to high concentrations of PFOA [215,216]. Moreover, Wang and colleagues found a positive relationship between maternal levels of PFOA and PFOS and IgE levels in the cord blood. Interestingly, the authors do not report any association between exposure levels and asthma symptoms when models are adjusted for other possible confounders [217]. However, it is relevant to report that several other studies demonstrated an inverse association between PFAS exposure and the risk of Eczema and Rhinitis [218]. 

Epidemiological studies in birth cohorts have shown that newborns exposed to high doses of PCBs in maternal blood, or children exposed to higher levels of PCBs in early childhood, have reduced thymus size, increased onset of respiratory disease, and reduced immune response following vaccination [219,220]. PCB exposure during pregnancy has been linked to abnormal cellular immunity development at 6 and 16 months after birth: subjects most exposed to PCBs had significantly higher expression of CD3^+^ T-lymphocytes, B-lymphocytes, and activated B-lymphocytes, while NK cells were less expressed. In the same study, altered serum immunoglobulins were found in subjects exposed to higher levels of PCBs in utero compared to those less exposed [221]. 

TCDD exposure during pregnancy has been observed to impact the miRNA profile of the fetal thymus, affecting the regulation of a wide number of genes that may impair immune system development [222]. TCDD also has a harmful effect on reproductive health, inducing epigenetic modifications such as DNA methylation in human germlines. This suggests that if TCDD exposure occurs during initial germ cell development, the alterations can be transmitted to subsequent generations [223]. Few studies have focused on TCDD’s multigenerational and transgenerational effects on human reproductive health, despite the quantity of evidence of such effects on male and female reproductive health in animal models. These studies show that paternal ancestral TCDD exposure substantially contributed to pregnancy outcomes and fetal health, although pregnancy outcomes were considered tightly related to the mother’s health [223].

Cases of occupational occurrence of allergic contact dermatitis in workers exposed to plastic based on BPA, and in some cases to bis-phenol F (BPF), have been reported, and for this reason, such compounds are classified as highly allergenic [224]. Several human birth cohort studies have reported an association between prenatal BPA exposure and allergy symptoms such as asthma and wheezing in children of different ages [225,226]. Although many articles mention allergenic effects as an additional way to demonstrate the immunotoxicity of BPA, there are currently few studies investigating the sensitization effects of BPA analogs [i.e., BPF and bisphenol S (BPS)]. Therefore, Mendyet and coworkers analyzed the NHANES data and found a positive correlation between urinary BPF levels and current asthma and hay fever, while BPS was associated with a higher likelihood of asthma in men [227]. 

In the United States, blood PBDE levels range from 30–100 ng/g of lipids in adults, but the alarming health concern was mainly based on children who showed blood PBDE levels 3- to 9-fold higher than adults. PBDEs disrupt the endocrine, immune, reproductive, and nervous systems. Studies performed on the Boston Birth Cohort provided evidence that in utero exposure to PBDEs may epigenetically reprogram the offspring’s immunological response through promoter methylation of a pro-inflammatory gene [228]. These data agree with the in vitro results using macrophage-like cell lines, where BDE-47 treatment modulated cytokine expression and secretion and miRNA expression. [97,98,99]. 

Finally, widening the field of investigation to include potential emerging pollutants, microplastics (MPs) are poised to become one of the next major challenges for research in the field of developmental immunotoxicology. As stated in a previous section of this manuscript, MPs are extraordinarily pervasive in the ecosystem, and their degradation occurs so slowly that a significant reduction in the coming decades seems improbable. Moreover, the use of plastic in current global production systems continues unabated. Single-use plastics constitute 35–40% of current plastic production and represent one of the fastest-growing segments in the plastics industry [229].

Starting from the research by Ragusa and colleagues, which first identified microplastics in the placenta, it has been suggested that these particles may affect the immune system [230]. Specifically, there is concern about their potential to alter maternal immune tolerance mechanisms critical to the success of the pregnancy. According to Zhu and coworkers, eleven different types of microplastics were identified in the placenta samples, with PVC (43.27%), PP (14.55%), and PBS (10.90%) being the three most widely represented types [231]. In addition, both Ragusa and Zhu also reported the presence of MPs in the fetal side of the placenta, suggesting that microplastics can reach the fetal bloodstream. More recently, Xue and colleagues detected MPs in 32 out of 40 samples of amniotic fluid [232]. Experimental studies in animal models have demonstrated that microplastics can interfere with the reproductive system and gametogenesis on multiple levels. These disruptions can range from hormonal imbalances to excessive ROS production in reproductive organs, ultimately affecting fertility and pregnancy outcomes [233]. Hu and colleagues demonstrated that exposure to PS MPs caused fetal losses in an allogeneic mouse mating model by altering the immune microenvironment. Their work highlighted changes in the production of pro/anti-inflammatory cytokines and alterations in the number and ratio of decidual natural killer cells and M1/M2 cells [234]. 

Although there is a growing body of research indicating the potential risks of exposure to microplastics, these findings are mostly derived from experiments involving animals rather than humans. Consequently, although these animal studies provide crucial information and raise significant concerns about the potential impacts of MPs on human health, there remains a need for more human-oriented research to fully understand the real magnitude of their harmful effects on humans.

It is becoming clear that DIT and prenatal programming may have a crucial role in raising the risk of infectious and non-communicable disease incidence; thus, greater efforts are needed to understand the events leading to DIT in order to effectively counteract them. 

An initial intervention could involve evaluating populations exposed to various concentration ranges by standardizing exposure variables and identifying priority outcomes for multicenter studies. Most of the studies mentioned in this paragraph examine the association between exposure and immune system outcomes in highly exposed populations. While these studies provide valuable insights into the effects of high levels of exposure, they do not fully reflect potential health risks. The immune system responses and health outcomes observed at high exposure levels might differ significantly from those at lower levels, potentially leading to an underestimation or overestimation of risks. For example, focusing on highly exposed populations might cause studies to miss subtler effects that occur at lower, more common exposure levels. Conversely, highly exposed populations also represent an important focus for studying co-exposures.

For instance, in a cohort of pregnant women living in a highly contaminated area, Longo and coworkers studied the effects of simultaneous exposure to a multi-pollutant mixture using a WQS regression model, demonstrating the association between the expression levels of immune-relevant miRNAs and levels of a suite of inorganic and organic elements in the sera of pregnant women. The study emphasized the concurrent assessment of essential elements, recognizing their indispensable role in maintaining physiological balance. This approach allowed the interplay between exposure to environmental pollutants and the availability of essential elements to be unraveled, shedding light on potential synergistic or antagonistic effects, with particular reference to epigenetic alterations inherent in the maintenance of the redox state and cellular homeostasis during pregnancy [235]. 

## 4. Conclusions

The intricate relationship between chemical pollution and immunity underscores the profound impact of environmental factors on human health. In an era of an increasingly polluted world where chemicals pervade our air, water, food, and soil, safeguarding the resilience of the human immune system is of paramount importance for the well-being of global populations. This situation and the associated risks are greatly amplified in areas where contaminants from industrial (i.e., heavy metals, persistent organic pollutants (POPs), etc.) and hazardous waste pollutants (microplastics, endocrine-disrupting chemicals, etc.) are released into the environment, leading to a complex mixture of substances determined to be harmful to human health. An estimation of the population living near landfills in the EU is about 30 million people, about 6% of the total European population, and it is estimated that there are 2.5 million contaminated sites in Europe, with potentially significant adverse health effects [236]. In this regard, the European Parliament has issued the REACH Regulation [Regulation on the Registration, Evaluation, Authorisation, and Restriction of Chemical Substances (EC) No. 1907/2006] to define the risks that may be posed by chemicals. Indeed, despite some impressive associations demonstrated by epidemiological studies on highly exposed populations, the role of pollutants on pathophysiological mechanisms responsible for the onset of diseases is often unknown. The REACH legislation provides toxicological studies in validated cell and animal model systems to assess the dose response to chemicals, the effects on molecular mechanisms, tissue damage, accumulation, and biomagnification under controlled dose and time exposures. The main outcome of these studies is to evaluate, for each pollutant, the No Observed Adverse Effect Level (NOAEL) and the Lowest Observed Adverse Effect Level (LOAEL), normally used to establish exposure threshold values in risk assessment guidelines. It is also noteworthy that the most recent studies in toxicology are focused on exposure to low-dose pollutant mixtures to better simulate the environmental conditions in real life. The early results suggest that low-dose mixtures could induce dysmetabolism and that pollutants could have a synergistic effect on systemic inflammatory responses, highlighting the need for policymakers to reassess the threshold values in risk assessment and improve strategies to protect human health [237,238,239]. Intervention strategies to alleviate the burden of immune system pathologies involve targeted environmental management and public health initiatives. In 2018, Bourguignon and coworkers proposed an approach aimed at reducing exposure to complex mixtures during pregnancy by implementing global initiatives to mitigate the possible consequences of combinations of adverse lifestyle factors. This theory, similar to the “Hygiene Theory” developed to decrease the burden of communicable diseases, involves reducing environmental exposures regardless of whether a causal link between exposure and illness has been identified. Indeed, considering that the effects of interactions among different pollutants on fetal and neonatal health are often unpredictable, and the long-term consequences are challenging to replicate experimentally, a strategy based on an *a priori* preventive approach could reduce the disease burden earlier compared to identifying causal links between exposure and illness [240]. Along these lines, Dietert and colleagues proposed another intervention strategy aimed at reducing the burden of immune system pathologies. According to the authors, a wide range of adult-onset chronic conditions could be prevented by an early diagnosis and intervention strategy for “entryway diseases” during the first years of life [162]. The identification of biomarkers of effect emerges as a potent tool for early disease detection for the understanding of the mechanisms underlying pollutant-induced pathologies. Indeed, advancements in scientific knowledge on the evidence-based link between environmental pollution and an altered immune response suggest the importance of (i) improving understanding of exposure risks; (ii) developing policy tools, guidelines, and recommendations for policymakers and authorities; and (iii) enhancing awareness and training for health professionals and citizens. Accurate and continuous information on this issue will promote general knowledge and an improvement of habits in the environmental health domain [241]. By implementing targeted interventions, we can pave the way towards a healthier and more resilient population, with a reduced burden of immune system pathologies. 

## Figures and Tables

**Figure 1 ijms-25-07788-f001:**
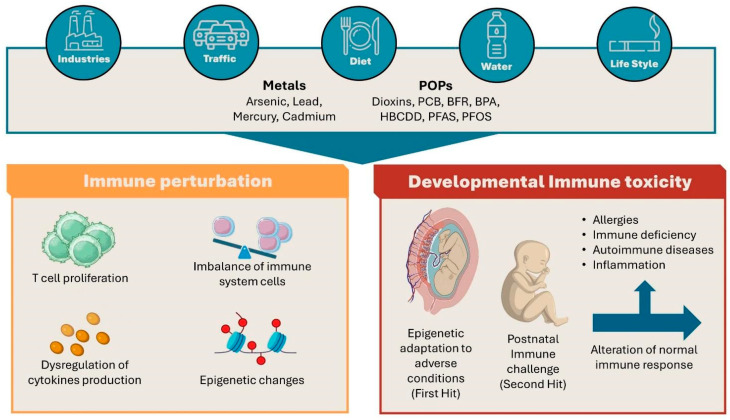
Schematic representation of the structure of the review.

**Figure 2 ijms-25-07788-f002:**
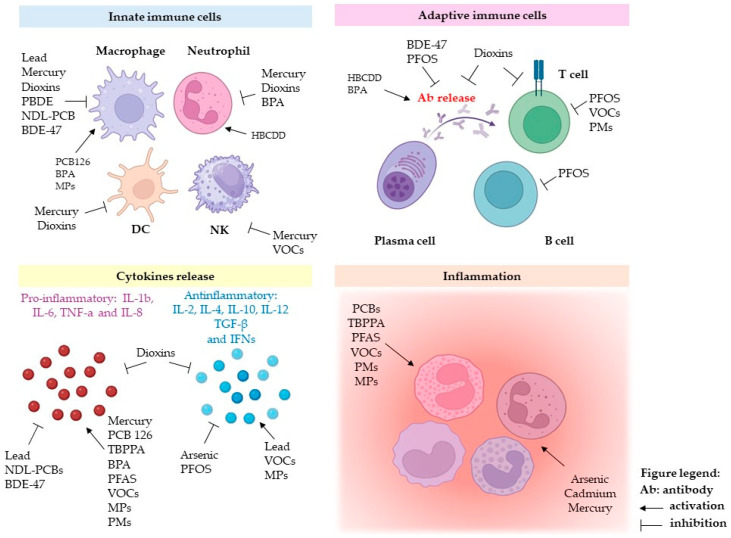
Schematic representation of the data obtained by in vitro, ex vivo, and in vivo studies on the effects of the main classes of environmental pollutants on immune response.

**Table 1 ijms-25-07788-t001:** In vitro, ex vivo, and in vivo effects of Heavy Metals.

Pollutant	Effects	Model Systems (In Vitro/Ex Vivo/In Vivo)	Reference
Arsenic	Reduction of (IFN-γ, IL-4, and IL-10)	Murine splenocytes, mouse activated T cell (ex vivo)	[34,35]
	Macrophages, neutrophil, B and T cell apoptosis induction	Human primary cultures (ex vivo)	[36]
	Modulation of T cell Receptor, the cell cycle, and apoptosis	Human PBMCs (ex vivo)	[37,38]
	Involvement in fibrinogenic processes and chronic hepatic fibrosis	Co-cultures of THP-1 macrophages or BMDM and primary lung fibroblasts from C57BL/6 mice (in vivo/in vitro/ex vivo); co-cultures of THP-1 monocytes and LX2 cell lines (in vitro)	[39,40]
	Hypomethylation of leukocyte DNA	Human blood samples (ex vivo)	[42]
	Oxidative stress signaling, DNA damage, and cytotoxicity in T cells and in human polymorphonuclear neutrophils	Human primary cell cultures and human cell lines (ex vivo, in vitro)	[44,45]
Lead	Induction of allergies and infectious and autoimmune diseases in humans	Human ex vivo studies	[46,47]
	Dysregulation of pro-inflammatory cytokines and impairment of THP-1 monocyte/macrophage cell viability	Human THP-1 Monocyte/Macrophage cultures (in vitro)	[48,49,50]
	Quiescence of hematopoietic stem cells	C57BL/6 murine model (in vivo and ex vivo)	[51]
	Induction of higher levels of IFN-γ, IL-2, IL-12, and IL-17 in exposed workers	Human THP-1 Monocytes/Macrophages (in vitro); Human serum samples and primary T cell cultures (ex vivo)	[52,53]
Mercury	Genetic damage and neurological, kidney, cardiac, and immunological diseases in humans	In vitro, ex vivo, and in vivo model systems	[56]
	Superoxide ion production and cytotoxic effect in neutrophils	Human neutrophils (ex vivo)	[57]
	Impairment of macrophage migratory and phagocytic activity; NO and pro-inflammatory cytokine production	BALB/cABOM peritoneal macrophages (ex vivo); Monocytes from Human PBMCs; co-cultures of Caco-2, HT29-MTX intestinal epithelial cells, and THP-1 macrophages (in vitro)	[58,59,60]
	Genotoxic effects. Suppression of both tumoricidal activity of blood and splenic NK cells and T and B cell proliferation	Human cell lines (in vitro) and blood samples (ex vivo); Balb/c mouse and rat primary cell cultures (in vivo and ex vivo)	[61,62,63]
Cadmium	Inflammation and oxidative damage induction in neutrophils and macrophages	Rat liver and kidney primary cell cultures (in vivo and ex vivo)*;* various cells and tissues in in vitro and ex vivo models.	[66,67]
	Downregulation of TNF-α, IL-12p40, TLRs, CD14, MD2, BD2, MyD88, p65, and NOS2	Wild boar macrophages (ex vivo)	[69]
	Impairment of adaptive immunity cell populations in offspring	C57Bl/6 mice (in vivo and ex vivo)	[70,71]
	Modulation of miRNAs associated with inflammation and carcinogenesis and alteration of Th17 and Treg lymphocyte subpopulations in exposed workers	Human blood samples (ex vivo)	[72]

**Table 2 ijms-25-07788-t002:** In vitro, ex vivo, and in vivo effects of POPs, VOCs, PMs, and microplastics (MPs).

Pollutant	Effects	Model System (In Vitro/Ex Vivo/In Vivo)	Reference
Dioxins	Impairment of macrophages, NK, neutrophils, and dendritic cells	In vivo murine models; in vitro and ex vivo cell cultures	[76,77,78]
	Reduction of both antibody production by B cells and the cytotoxic activity of T lymphocytes	In vitro and ex vivo cell cultures; in vivo murine models	[79]
	Attenuation of IgE-mediated hypersensitivity response	In vivo and ex vivo studies	[80]
	Alterations in THP-1 macrophage adherence, adhesion molecule expression, morphology, multiple cytokine/chemokine production, and total mRNA expression	THP-1 monocyte/macrophage cell line (in vitro)	[81]
	Effects on human reproductive health	Murine models; Human ex vivo studies	[82]
PCBs	Pro-inflammatory activity in macrophages	THP-1 monocyte/macrophage cell line (in vitro)	[83]
	In vitro immunosuppressive effects and expression of reactive species	J774A.1 cell line and primary murine macrophages (in vitro and ex vivo)	[84]
	In vivo inflammatory effects and impairment of immune system functions in a mouse model	Wild-type mice C57BL/6J and SVJ129 (in vivo and ex vivo)	[86]
	DNA methylation differences in PCB-exposed populations	Human PBMCs (ex vivo)	[87]
PBDEs	Liver, kidney, gut, and thyroid toxicity	In vitro, ex vivo, and in vivo studies	[91]
	Neurotoxic, cardiotoxic, hepatotoxic, and teratogenic effects on zebrafish and fish	Zebrafish embryos (in vivo and ex vivo)	[92,93,94]
	Impairment of pro-inflammatory response modulation of small extracellular vesicle biogenesis and miRNA cargo and exacerbation of LPS-induced pro-inflammatory response in a macrophage cell line	THP-1 monocyte/macrophage cell line (in vitro)	[95,96,97]
	Alteration of tight junctions, adhesion molecules, cytokines, and EMT (epithelial–mesenchymal transition) marker expression in epithelial lung cells	ALI cultures of human A549 cell line (in vitro)	[98]
	Cardiovascular toxicity	THP-1 monocyte/macrophage cell line (in vitro)	[99]
	Destruction of macrophage functional activity in animal model systems	RTG-2 cell line (in vitro)	[100,101,102]
	Reduction of antibody response and histopathological effects on the liver, spleen, small intestine, and thyroid	BALB/c murine model (in vivo and ex vivo)	[103]
HBCDD	Bioaccumulation in human blood, adipose tissue, and breast milk	Human blood, adipose tissue, and maternal milk (ex vivo)	[109,110,111]
TBBPA	Genotoxic effects	Human PBMCs (ex vivo)	[112,113,114]
	Induction of inflammatory phenotype in human dendritic cells from healthy subjects	Human monocyte-derived dendritic cells (ex vivo)	[115]
	Neurotoxic, nephrotoxic, hepatotoxic, and immunotoxic effects		[116]
	Correlation with human thyroid and neurological disorders, reproductive health, immunological, oncological, and cardiovascular diseases	Laboratory animals and human samples (in vivo and ex vivo)	[117,118]
BPA	Impairment of chemotactic function in neutrophils	Neutrophils isolated from human blood	[119]
	Induction of polarization in M1 macrophages	Peritoneal macrophages from C57BL/6 J mice (ex vivo)	[120]
	Upregulation of pro-inflammatory cytokines	RAW264.7 murine macrophage cell line (in vitro)	[121,122]
	Immunotoxic effects on adaptive immune response and effect on IgM reactivity against tumor antigens	Guinea pig model system (in vivo and ex vivo)	[123]
	Inhibition of T cell proliferation	BALB/c murine model (in vivo and ex vivo)	[102]
PFAS (PFOS and PFOA)	PFOS bioaccumulation in human liver, kidneys, lungs, hair, breast milk, urine, and blood	Human blood, urine, milk, hair, nail, and tissue samples	[125,126]
	In vitro reduction of IL-2 expression in human T cells	Jurkat cell line and primary T cells (in vitro and ex vivo)	[127]
	De novo localization of immune cell populations in organs such as the spleen and liver	C57BL/6 murine model (in vivo and ex vivo)	[128]
	Reduction of B-cell subtypes and IgM antibody primary response in female mice induced by PFOA	C57BL/6 murine model (in vivo and ex vivo)	[129]
VOCs	Increased TNF-α expression by toluene	CD3+/CD28+ human PBMCs (ex vivo)	[133]
	Induction of oxidative stress and mitochondrial damage by xilene	Human T Lymphocytes (ex vivo)	[134]
	Induction of apoptosis pathways by means of INF signal cells	Human promyelocytic leukemia HL60 cells (in vitro)	[135]
	Reduction of NK, T regulatory, and CD8+ effector T cells	PBMCs from workers’ cohort (ex vivo)	[136]
PMs	Lung tissue damage and impairment of innate immune responses induced by a high concentration of PM2.5	Rat model (in vivo)	[139]
	Induction of inflammatory response, upregulation of cytokines and chemokines; downregulation of pathogen defense and antiviral factors by PM10	Human PBMC cells (ex vivo)	[140]
	Induction of pro-inflammatory response and reduction of antiviral defenses by ultrafine particulate	ALI cultures of Calu-3 cell line (in vitro)	[145]
	Activation of pro-inflammatory response by PM	ALI cultures of A549 and monocytes (in vitro)	[146]
	Impairment of regulatory T cells (Treg) and disruption of T-helper 1 (Th1) and T-helper 2 (Th2) cell balance by PM 2.5	Rat model (in vivo)	[147]
MPs	Alteration of cellular morphology and modulation of IL-1β, CCL2, and TGF-β levels by PS MPs	Microglial HMC-3 cell	[154]
	Leucocyte impairment and activation of oxidative stress by PVC or PE	Sea bass or seabream fish (in vitro)	[155]
	Liver damage, inflammation, autophagy, intestinal macrophage infiltration, and lipid accumulation by PS MPs	Chicken models (in vivo)	[156]
	Induction of pro-inflammatory cytokines and histamine release by PP particles	PBMCs, Raw 264.7, and HMC-1 cells (in vitro and ex vivo)	[157]
	Cytotoxic activities, production of ROS, and macrophage polarization by PS MPs	PBMC and THP-1 macrophage cells lines	[158]

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
