# Peer review of "Guardians under Siege: Exploring Pollution’s Effects on Human Immunity"

_ijms, 2024, doi:10.3390/ijms25147788_

Round 1

Reviewer 1 Report (New Reviewer)

Comments and Suggestions for Authors

The manuscript is well written, and covers the field of research well and is mostly valid for publication, the abstract and organisation of the manuscript is clear. Congratulations to the authors, but I have a few comments to improve the quality of the manuscript.

Firstly, the authors could add a paragraph explaining microplastics pollution, highlighting on why microplastics pollution are of such a huge impact on human health and pregnancy. Next, the authors could add a paragraph on VOCs (volatile organic compounds), which would further strengthen the quality of this manuscript.
Finally, this is a minor suggestions, but in the figures, the authors should use the sans serif fonts (i.e. Arial), to make it neater.

Author Response

The manuscript is well written, and covers the field of research well and is mostly valid for publication, the abstract and organisation of the manuscript is clear. Congratulations to the authors, but I have a few comments to improve the quality of the manuscript.

We are grateful to the Reviewer for Her/His interest in our work.

Firstly, the authors could add a paragraph explaining microplastics pollution, highlighting on why microplastics pollution are of such a huge impact on human health and pregnancy. Next, the authors could add a paragraph on VOCs (volatile organic compounds), which would further strengthen the quality of this manuscript.

We thank the Reviewer for their suggestions. The topics suggested by the Reviewer are very relevant in the field and likely, due to the large set of data, deserve a review in their own right. However, to follow the advice, additional paragraphs have been introduced.

Finally, this is a minor suggestions, but in the figures, the authors should use the sans serif fonts (i.e. Arial), to make it neater.

The point was addressed

Reviewer 2 Report (New Reviewer)

Comments and Suggestions for Authors

The review submitted by Drago and colleagues, is a comprehensive review about the impact of xenobiotics on the immune system.  The review is well written and could be of great interest to the IJMS audience. Please see below my comments

When possible,  please provide molecular mechanisms or modes of action of the discussed contaminant. ie.

1) lines 558-564  is this hypo or hypermethylation?, which genes are susceptible etc.

2) lines 407-411. How does BPA-induced ROS affect the ability of neutrophils to eliminate pathogens?

Could the authors discuss in detail the effects of particulate matter on the immune system. It is mentioned briefly in the introduction.  PM is ubiquitous and it is well-known to have effects on the immune system.

Would the authors consider including a section on emerging contaminants such as microplastics and nanomaterials? I think this could help to increase the impact of the review.

Comments on the Quality of English Language

Please proofread the manuscript to avoid some minor grammar errors

Author Response

The review submitted by Drago and colleagues, is a comprehensive review about the impact of xenobiotics on the immune system.  The review is well written and could be of great interest to the IJMS audience. Please see below my comments

We are grateful to the Reviewer for Her/His interest in our work.

When possible,  please provide molecular mechanisms or modes of action of the discussed contaminant. ie.

1) lines 558-564  is this hypo or hypermethylation?, which genes are susceptible etc.

2) lines 407-411. How does BPA-induced ROS affect the ability of neutrophils to eliminate pathogens?

We thank the reviewer for the suggestion and understand the reasoning behind it. Following the reviewer's suggestions, additional data have been introduced into the manuscript. However, we believe that providing a detailed description of the molecular mechanisms of action for all the pollutants mentioned in this manuscript would be very challenging. A more detailed, ad hoc review on each class of xenobiotic would likely address this point more effectively.

Could the authors discuss in detail the effects of particulate matter on the immune system. It is mentioned briefly in the introduction.  PM is ubiquitous and it is well-known to have effects on the immune system.

We thank we the Reviewer for Her/His suggestions. A new set of information has been added to the manuscript to address the suggestion

Would the authors consider including a section on emerging contaminants such as microplastics and nanomaterials? I think this could help to increase the impact of the review.

A new set of information has been added to the manuscript regarding microplastics

Comments on the Quality of English Language

Please proofread the manuscript to avoid some minor grammar errors

Language has been reviewed

This manuscript is a resubmission of an earlier submission. The following is a list of the peer review reports and author responses from that submission.

Round 1

Reviewer 1 Report

Comments and Suggestions for Authors

Whilst various reviews have been published on the topic on the impact of environmental pollutants on human health, this paper takes on an interesting and specific view on the immunity in adult life as well as during pre- and post-natal development.

While the narrative is interesting and the paper reads well, I do have several comments that could be incorporated into the manuscript:

- when describing the specific effects each pollutant has on the immune system, please specify if the observations are gathered from an in vivo/animal/in vitro study. Perhaps it would be of value to divide each subsection related to a specific pollutant into in vivo and in vitro effects. 

- The manuscript seems to end rather abruptly. A summary or take home message would provide a proper review of the topic to the reader. The authors could also compare the similarities and differences in the effects heavy metals, BPA, dioxins or PCB exhibit on the immune system.

- It may be beneficial to introduce each pollutant in 3-4 sentences on a more general level, similarly to how Mercury was introduced.

- The authors could add a section on current strategies to prevent or minimise the adverse effects environmental contaminants on the immune system. Also, what are current management or treatment strategies to alleviate environmental immunotoxicity?

Author Response

Dear Editor,

First of all, we would like to thank the Reviewers for their suggestions to improve our manuscript.

The manuscript “Guardians Under Siege: Exploring Pollution's effects on Human Immunity” has been revised according to the suggestions and comments of the Reviewers. Specifically, we have integrated the text with general information about the different pollutants and improved the figures and tables, to clarify the effect of the different pollutants on immune response in in vitro, ex vivo and in vivo studies. Furthermore, thanks to the reviewers’ suggestions, we have   the “Conclusions” paragraphs to explore the topic on the current strategies to minimise the adverse effects of environmental contaminants.

Therefore, we report below a point-by-point reply to the comments.

Reviewer 1-Comments and Suggestions for Authors

Whilst various reviews have been published on the topic on the impact of environmental pollutants on human health, this paper takes on an interesting and specific view on the immunity in adult life as well as during pre- and post-natal development.

While the narrative is interesting and the paper reads well, I do have several comments that could be incorporated into the manuscript:

  1. when describing the specific effects each pollutant has on the immune system, please specify if the observations are gathered from an in vivo/animal/in vitro study. Perhaps it would be of value to divide each subsection related to a specific pollutant into in vivo and in vitro effects.

We thank the Reviewer for this useful suggestion. To clarify whether the data were obtained from in vitro/ex vivo or in vivo studies, we have modified tables 1 and 2 by adding the column “model system”, in which we introduced the   information about the experimental model.  

  1. The manuscript seems to end rather abruptly. A summary or take-home message would provide a proper review of the topic to the reader. The authors could also compare the similarities and differences in the effects heavy metals, BPA, dioxins or PCB exhibit on the immune system.

We particularly thank the Reviewer for this very relevant comment. To highlight similarities and differences among the effects of different pollutants or classes of pollutants on immune response, we have introduced a new figure in the manuscript (figure 2). Figure 2 is a schematic representation of the effect of environmental pollutants on immune cell subtypes (innate or adaptive cells) and mechanisms of immune modulation as well as cytokines release and inflammation. Furthermore, we revised the manuscript conclusions as requested by the reviewer.

  1. It may be beneficial to introduce each pollutant in 3-4 sentences on a more general level, similarly to how Mercury was introduced.

We thank the Reviewer for the comment. We have modified the text of each subsection to provide general information about the specific pollutants.

  1. The authors could add a section on current strategies to prevent or minimise the adverse effects environmental contaminants on the immune system. Also, what are current management or treatment strategies to alleviate environmental immunotoxicity  

         We particularly thank the Reviewer for this suggestion. In the revised manuscript we have explored recent studies regarding the current strategies to minimize the adverse effect of pollutant on immune response in the “DIT” (lines 648-658 and 673-676) and “Conclusion” sections. Finally, we have integrated the pertinent  publications in the references section.

Reviewer 2 Report

Comments and Suggestions for Authors

Based on the concept that the immune system may be affected by environmental pollutants the authors have assembled a great number of in vitroex vivo and in vivo studies to explore potential health effects in humans. Unfortunately, there is no evaluation to what extent the studies are relevant for humans. This would include an extrapolation of the effective concentrations of the in vitro studies to human exposure as well as an evaluation of the reliability of epidemiological studies of exposed individuals. Without this the authors just provide an accumulation of data without evaluation, whether the data are relevant to humans and supported by reliable observations in humans at specific exposures. Without this the manuscript is of no added value. 

Comments on the Quality of English Language

Minor editing only.

Author Response

Dear Editor,

First of all, we would like to thank the Reviewers for their suggestions to improve our manuscript.

The manuscript “Guardians Under Siege: Exploring Pollution's effects on Human Immunity” has been revised according to the suggestions and comments of the Reviewers. Specifically, we have integrated the text with general information about the different pollutants and improved the figures and tables, to clarify the effect of the different pollutants on immune response in in vitro, ex vivo and in vivo studies. Furthermore, thanks to the reviewers’ suggestions, we have   the “Conclusions” paragraphs to explore the topic on the current strategies to minimise the adverse effects of environmental contaminants.

Therefore, we report below a point-by-point reply to the comments.

Reviewer 2-Comments and Suggestions for Authors

Based on the concept that the immune system may be affected by environmental pollutants the authors have assembled a great number of in vitroex vivo and in vivo studies to explore potential health effects in humans. Unfortunately, there is no evaluation to what extent the studies are relevant for humans. This would include an extrapolation of the effective concentrations of the in vitro studies to human exposure as well as an evaluation of the reliability of epidemiological studies of exposed individuals. Without this the authors just provide an accumulation of data without evaluation, whether the data are relevant to humans and supported by reliable observations in humans at specific exposures. Without this the manuscript is of no added value

Thanks for the comment. In this review, we have described the main knowledge related to the effects of environmental pollutants on immune response, focusing our attention both on experimental models (in vitro, ex vivo and in vivo) and studies on general populations, cohort of workers and birth cohorts. We respectfully believe that the description of laboratory experimental data can be relevant to understand the toxicology pathway involved by the pollution treatment. Mechanistic biological studies can be essential for understanding the cytotoxic and genotoxic effects induced by environmental pollutants that cannot be easily studied through epidemiological approaches. That is the reason why we decided to design our review with two sections regarding the biological studies and the “real life” epidemiological studies. We have provided an analysis, ( studies on cohorts of populations), offering insight into the impact of contaminants on human health and the development of various diseases associated with immune responses.      Furthermore, in the new revised form     we improved the tables and the figures and we modified the conclusions section suggesting the current strategies to minimize the adverse effect of pollutants on immune response.

Round 2

Reviewer 2 Report

Comments and Suggestions for Authors

Apparently, the authors did not understand the reviewer’s point of critcism, because as before they did not evaluate the human relevance of the accumulated data base presented. A science based review would need a descrption of doses used in the different experiments as compared to possible human exposure. Similarly, relevance of in vitro studies can only be evaluated by comparing the concentrations used in the experiments with concentartions in the blood or at relevant targets in exposed humans. Without such information the manuscript is of no added value.

Comments on the Quality of English Language

no comments